# PROGPROMPT: Generating Situated Robot Task Plans using Large Language Models

Ishika Singh[1], Valts Blukis[2], Arsalan Mousavian[2], Ankit Goyal[2], Danfei Xu[2],
Jonathan Tremblay[2], Dieter Fox[2], Jesse Thomason[1], Animesh Garg[2]

*Abstract*— Task planning can require defining myriad domain knowledge about the world in which a robot needs to act. To ameliorate that effort, large language models (LLMs) can be used to score potential next actions during task planning, and even generate action sequences directly, given an instruction in natural language with no additional domain information. However, such methods either require enumerating all possible next steps for scoring, or generate free-form text that may contain actions not possible on a given robot in its current context. We present a programmatic LLM prompt structure that enables plan generation functional across situated environments, robot capabilities, and tasks. Our key insight is to prompt the LLM with program-like specifications of the available actions and objects in an environment, as well as with example **programs** that can be executed. We make concrete recommendations about prompt structure and generation constraints through ablation experiments, demonstrate state of the art success rates in VirtualHome household tasks, and deploy our method on a physical robot arm for tabletop tasks. Website at progprompt.github.io

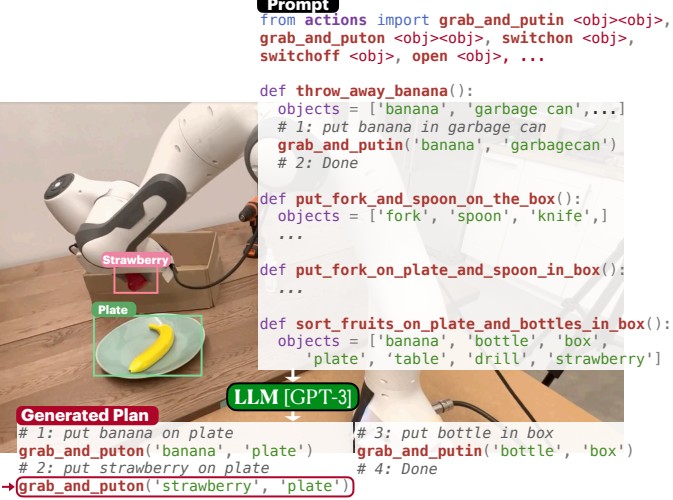

**Fig. 1:** PROGPROMPT leverages LLMs' strengths in both world knowledge and programming language understanding to generate situated task plans that can be directly executed.

## I. INTRODUCTION

Everyday household tasks require *both* commonsense understanding of the world and situated knowledge about the current environment. To create a task plan for "Make dinner," an agent needs common sense: *object affordances*, such as that the stove and microwave can be used for heating; *logical sequences of actions*, such as an oven must be preheated before food is added; and *task relevance of objects and actions*, such as heating and food are actions related to "dinner" in the first place. However, this reasoning is infeasible without *state feedback*. The agent needs to know what food is available *in the current environment*, such as whether the freezer contains fish or the fridge contains chicken.

Autoregressive large language models (LLMs) trained on large corpora to *generate* text sequences conditioned on input *prompts* have remarkable multi-task generalization. This ability has recently been leveraged to generate plausible action plans in context of robotic task planning [1], [2], [3], [4] by either scoring next steps or generating new steps directly. In scoring mode, the LLM evaluates an enumeration of actions and their arguments from the space of what's possible. For instance, given a goal to "Make dinner" with first action being "open the fridge", the LLM could score a list of possible actions: "pick up the chicken", "pick up the soda", "close the fridge", ..., "turn on the lightswitch." In text-generation mode, the LLM can produce the next few

Correspondence to: ishikasi@usc.edu
This work was done while IS was an intern at NVIDIA
[1]University of Southern California, [2]NVIDIA

words, which then need to be mapped to actions and world objects available to the agent. For example, if the LLM produced "reach in and pick up the jar of pickles," that string would have to neatly map to an executable action like "pick up jar." A key component missing in LLM-based task planning is state feedback from the environment. The fridge in the house might not contain chicken, soda, or pickles, but a high-level instruction "Make dinner" doesn't give us that world state information. **Our work introduces situated-awareness in LLM-based robot task planning.**

We introduce PROGPROMPT, a prompting scheme that goes beyond conditioning LLMs in natural language. PROGPROMPT utilizes *programming language* structures, leveraging the fact that LLMs are trained on vast web corpora that includes many programming tutorials and code documentation (Fig. 1). PROGPROMPT provides an LLM a Pythonic program header that imports available actions and their expected parameters, shows a list of environment objects, and then defines functions like make_dinner whose bodies are sequences of actions operating on objects. We incorporate situated state feedback from the environment by asserting preconditions of our plan, such as being close to the fridge before attempting to open it, and responding to failed assertions with recovery actions. What's more, we show that including natural language *comments* in PROGPROMPT programs to explain the goal of the upcoming action improves task success of generated plan programs.

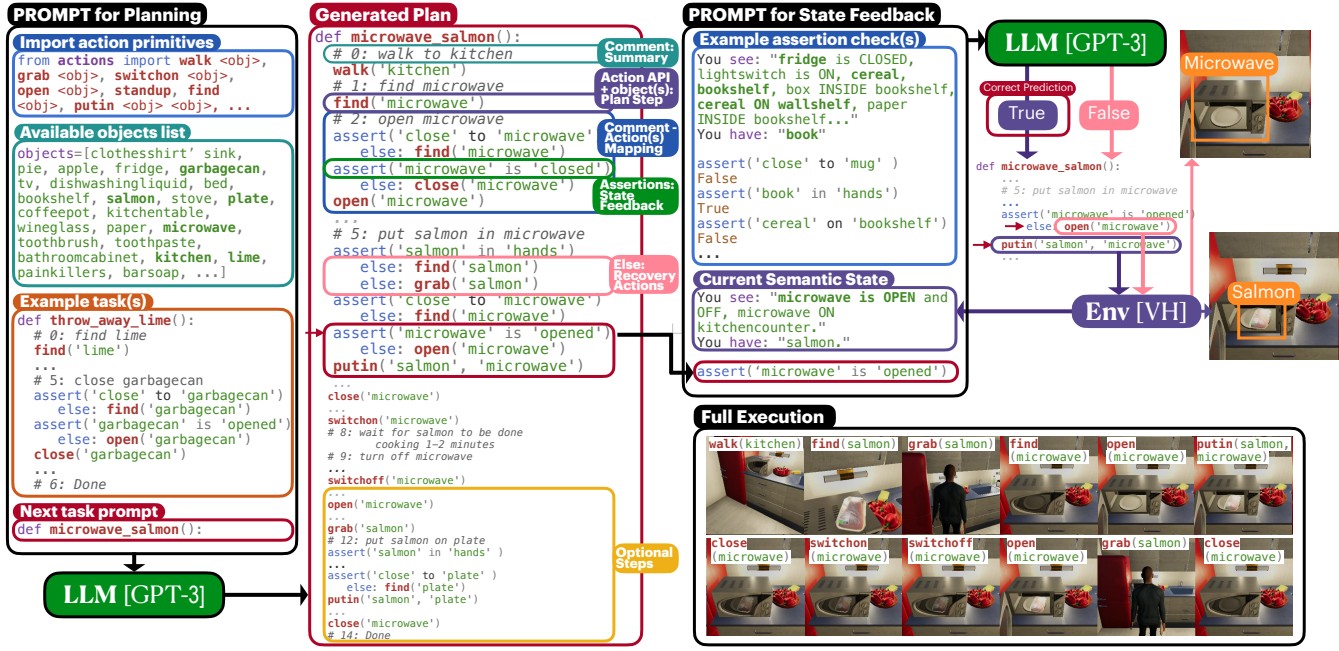

**Fig. 2:** Our PROGPROMPTs include import statement, object list, and example tasks (*PROMPT for Planning*). The *Generated Plan* is for microwave salmon. We highlight prompt comments, actions as imported function calls with objects as arguments, and assertions with recovery steps. *PROMPT for State Feedback* represents example assertion checks. We further show execution of the program. We illustrate a scenario where an assertion succeeds or fails, and how the generated plan corrects the error before executing the next step. *Full Execution* of the program is shown in bottom-right.

## II. BACKGROUND AND RELATED WORK

**Task Planning.** For high-level planning, most works in robotics use search in a pre-defined domain [5], [6], [7]. Unconditional search can be hard to scale in environments with many feasible actions and objects [8], [9] due to large branching factors. Heuristics are often used to guide the search [10], [11], [12], [13]. Recent works have explored learning-based task & motion planning, using methods such as representation learning, hierarchical learning, language as planning space, learning compositional skills and more [14], [15], [16], [17], [18], [19], [20], [21], [22], [23], [24], [25], [26]. Our method *sidesteps* search to directly generate a plan that includes conditional reasoning and error-correction.

We formulate task planning as the tuple $\langle \mathcal{O}, \mathcal{P}, \mathcal{A}, \mathcal{T}, \mathcal{I}, \mathcal{G}, \mathbf{t} \rangle$. $\mathcal{O}$ is a set of all the objects available in the environment, $\mathcal{P}$ is a set of properties of the objects which also informs object affordances, $\mathcal{A}$ is a set of executable actions that changes depending on the current environment state defined as $\mathbf{s} \in \mathcal{S}$. A state $\mathbf{s}$ is a specific assignment of all object properties, and $\mathcal{S}$ is a set of all possible assignments. $\mathcal{T}$ represents the transition model $\mathcal{T} : \mathcal{S} \times \mathcal{A} \to \mathcal{S}$, $\mathcal{I}$ and $\mathcal{G}$ are the initial and goal states. The agent does not have access to the goal state $\mathbf{g} \in \mathcal{G}$, but only a high-level task description $\mathbf{t}$.

Consider the task $\mathbf{t} =$ "*microwave salmon*". Task relevant objects *microwave, salmon* $\in \mathcal{O}$ will have properties modified during action execution. For example, action $\mathbf{a} = \texttt{open}(\textit{microwave})$ will change the state from $\texttt{closed}(\textit{microwave}) \in \mathbf{s}$ to $\neg\texttt{closed}(\textit{microwave}) \in \mathbf{s}'$ if $\mathbf{a}$ is admissible, i.e., $\exists(\mathbf{a}, \mathbf{s}, \mathbf{s}')$ $s.t.$ $\mathbf{a} \in \mathcal{A} \wedge \mathbf{s}, \mathbf{s}' \in \mathcal{S} \wedge \mathcal{T}(\mathbf{s}, \mathbf{a}) = \mathbf{s}'$. In this exam-

ple a goal state $\mathbf{g} \in \mathcal{G}$ could contain the conditions $\texttt{heated}(\textit{salmon}) \in \mathbf{g}$, $\neg\texttt{closed}(\textit{microwave}) \in \mathbf{g}$ and $\neg\texttt{switchedOn}(\textit{microwave}) \in \mathbf{g}$.

**Planning with LLMs.** A Large Language Model (LLM) is a neural network with many parameters—currently hundreds of billions [27], [28]—trained on unsupervised learning objectives such as next-token prediction or masked-language modelling. An autoregressive LLM is trained with a maximum likelihood loss to model the probability of a sequence of tokens $\mathbf{y}$ conditioned on an input sequence $\mathbf{x}$, i.e. $\theta = \arg\max_\theta P(\mathbf{y}|\mathbf{x}; \theta)$, where $\theta$ are model parameters. The trained LLM is then used for prediction $\hat{\mathbf{y}} = \arg\max_{\mathbf{y} \in \mathbb{S}} P(\mathbf{y}|\mathbf{x}; \theta)$, where $\mathbb{S}$ is the set of all text sequences. Since search space $\mathbb{S}$ is huge, approximate decoding strategies are used for tractability [29], [30], [31].

LLMs are trained on large text corpora, and exhibit multi-task generalization when provided with a relevant prompt input $\mathbf{x}$. Prompting LLMs to generate text useful for robot task planning is a nascent topic [32], [33], [34], [2], [4], [1]. Prompt design is challenging given the lack of paired natural language instruction text with executable plans or robot action sequences [35]. Devising a prompt for task plan prediction can be broken down into a *prompting function* and an *answer search* strategy [35]. A *prompting function*, $f_{\text{prompt}}(.)$ transforms the input state observation $\mathbf{s}$ into a textual prompt. *Answer search* is the generation step, in which the LLM outputs from the entire LLM vocabulary or scores a predefined set of options.

Closest to our work, [2] generates open-domain plans using LLMs. In that work, planning proceeds by: 1) selecting a similar task in the prompt example ($f_{\text{prompt}}$); 2) open-

```
def put_salmon_in_microwave():
    # 1: grab salmon
    assert('close' to 'salmon')
        else: find('salmon')
    grab('salmon')
    # 2: put salmon in microwave
    assert('salmon' in 'hands' )
        else: find('salmon')
        else: grab('salmon')
    assert('close' to 'microwave' )
        else: find('microwave')
    assert('microwave' is 'opened')
        else: open('microwave')
    putin('salmon', 'microwave')
```

**Fig. 3:** Pythonic PROGPROMPT plan for "*put salmon in the microwave.*"

ended task plan generation (answer search); and 3) 1:1 prediction to action matching. The entire plan is generated *open-loop without any environment interaction*, and later tested for executability of matched actions. However, action matching based on generated text doesn't ensure the action is admissible in the current situation. INNERMONOLOGUE [1] introduces environment feedback and state monitoring, but still found that LLM planners proposed actions involving objects not present in the scene. Our work shows that a *programming language-inspired prompt generator* can inform the LLM of both situated environment state and available robot actions, ensuring output compatibility to robot actions.

The related SAYCAN [4] uses natural language prompting with LLMs to generate a set of feasible planning steps, rescoring matched admissible actions using a learned value function. SayCan constructs a set of all admissible actions expressed in natural language and scores them using an LLM. This is challenging to do in environments with combinatorial action spaces. Concurrent with our work are Socratic models [3], which also use code-completion to generate robot plans. We go beyond [3] by leveraging additional, familiar features of programming languages in our prompts. We define an $f_{prompt}$ that includes import statements to model robot capabilities, natural language comments to elicit common sense reasoning, and assertions to track execution state. Our answer search is performed by allowing the LLM to generate an entire, executable plan program directly.

## III. OUR METHOD: PROGPROMPT

We represent robot plans as *pythonic* programs. Following the paradigm of LLM prompting, we create a prompt structured as pythonic code and use an LLM to complete the code (Fig. 2). We use features available in Python to construct prompts that elicit an LLM to generate situated robot task plans, conditioned on a natural language instruction.

### A. Representing Robot Plans as Pythonic Functions

Plan functions consist of API calls to action primitives, comments to summarize actions, and assertions for tracking execution (Fig. 3). Primitive actions use objects as arguments. For example, the "*put salmon in the microwave*" task includes API calls like find(*salmon*).

We utilize comments in the code to provide natural language summaries for subsequent sequences of actions. Com-

ments help break down the high-level task into logical subtasks. For example, in Fig. 3, the "*put salmon in microwave*" task is broken down into sub-tasks using comments "# grab salmon" and "# put salmon in microwave". This partitioning could help the LLM to express its knowledge about tasks and sub-tasks in natural language and aid planning. Comments also inform the LLM about immediate goals, reducing the possibility of incoherent, divergent, or repetitive outputs. Prior work [36] has also shown the efficacy of similar intermediate summaries called 'chain of thought' for improving performance of LLMs on a range of arithmetic, commonsense, and symbolic reasoning tasks. We empirically verify the utility of comments (Tab. I; column COMMENTS).

Assertions provide an environment feedback mechanism to make sure that the preconditions hold, and enable error recovery when they do not. For example, in Fig. 3, before the grab(*salmon*) action, the plan asserts the agent is close to *salmon*. If not, the agent first executes find(*salmon*). In Tab. I, we show that such assert statements (column FEEDBACK) benefit plan generation.

### B. Constructing Programming Language Prompts

We provide information about the environment and primitive actions to the LLM through prompt construction. As done in few-shot LLM prompting, we also provide the LLM with examples of sample tasks and plans. Fig. 2 illustrates our prompt function $f_{prompt}$ which takes in all the information (observations, action primitives, examples) and produces a Pythonic prompt for the LLM to complete. The LLM then predicts the <next_task>(.) as an executable function (microwave_salmon in Fig. 2).

In the task microwave_salmon, a reasonable first step that an LLM could generate is take_out(*salmon, grocery bag*). However, the agent responsible for the executing the plan might not have a primitive action to take_out. To inform the LLM about the agent's action primitives, we provide them as Pythonic import statements. These encourage the LLM to restrict its output to only functions that are available in the current context. To change agents, PROGPROMPT just needs a new list of imported functions representing agent actions. A *grocery bag* object might also not exist in the environment. We provide the available objects in the environment as a list of strings. Since our prompting scheme explicitly lists out the set of functions and objects available to the model, the generated plans typically contain actions an agent can take and objects available in the environment.

PROGPROMPT also includes a few example tasks—fully executable program plans. Each example task demonstrates how to complete a given task using available actions and objects in the given environment. These examples demonstrate the relationship between task name, given as the function handle, and actions to take, as well as the restrictions on actions and objects to involve.

### C. Task Plan Generation and Execution

The given task is fully inferred by the LLM based on the PROGPROMPT prompt. Generated plans are executed on a

**TABLE I:** Evaluation of generated programs on Virtual Home. PROGPROMPT uses 3 fixed example programs, except the DAVINCI backbone which can fit only 2 in the available API. [2] use 1 dynamically selected example, as described in their paper. LANGPROMPT uses 3 natural language text examples. Best performing model with a GPT3 backbone is shown in blue (used for ablation studies); best performing model overall shown in **bold**. PROGPROMPT significantly outperforms the baseline [2] and LANGPROMPT. We also showcase how each PROGPROMPT feature adds to the performance of the method.

| # | — Prompt Format and Parameters — | | | LLM Backbone | SR | Exec | GCR |
|---|---|---|---|---|---|---|---|
| | Format | COMMENTS | FEEDBACK | | | | |
| 1 | PROGPROMPT | ✓ | ✓ | CODEX | **0.40**±0.11 | **0.90**±0.05 | **0.72**±0.09 |
| 2 | PROGPROMPT | ✓ | ✓ | DAVINCI | 0.22±0.04 | 0.60±0.04 | 0.46±0.04 |
| 3 | PROGPROMPT | ✓ | ✓ | GPT3 | 0.34±0.08 | 0.84±0.01 | 0.65±0.05 |
| 4 | PROGPROMPT | ✓ | ✗ | GPT3 | 0.28±0.04 | 0.82±0.01 | 0.56±0.02 |
| 5 | PROGPROMPT | ✗ | ✓ | GPT3 | 0.30±0.04 | 0.65±0.01 | 0.58±0.02 |
| 6 | PROGPROMPT | ✗ | ✗ | GPT3 | 0.18±0.04 | 0.68±0.01 | 0.42±0.02 |
| 7 | LANGPROMPT | - | - | GPT3 | 0.00±0.00 | 0.36±0.00 | 0.42±0.02 |
| 8 | Baseline from HUANG ET AL. [2] | | | GPT3 | 0.00±0.00 | 0.45±0.03 | 0.21±0.03 |

virtual agent or a physical robot system using an interpreter that executes each action command against the environment. Assertion checking is done in a closed-loop manner during execution, providing current environment state feedback.

## IV. EXPERIMENTS

We evaluate our method with experiments in a virtual household environment and on a physical robot manipulator.

### A. Simulation Experiments

We evaluate our method in the Virtual Home (VH) Environment [8], a deterministic simulation platform for typical household activities. A VH state $\mathbf{s}$ is a set of objects $\mathcal{O}$ and properties $\mathcal{P}$. $\mathcal{P}$ encodes information like in(*salmon, microwave*) and agent_close_to(*salmon*). The action space is $\mathcal{A} = \{$grab, putin, putback, walk, find, open, close, switchon, switchoff, sit, standup$\}$.

We experiment with 3 VH environments. Each environment contains 115 unique object instances (Fig. 2), including class-level duplicates. Each object has properties corresponding to its action affordances. Some objects also have a semantic state like heated, washed, or used. For example, an object in the *Food* category can become heated whenever in(*object, microwave*) ∧ switched_on(*microwave*).

We create a dataset of 70 household tasks. Tasks are posed with high-level instructions like "*microwave salmon*". We collect a ground-truth sequence of actions that completes the task from an initial state, and record the final state $\mathbf{g}$ that defines a set of symbolic goal conditions, $\mathbf{g} \in \mathcal{P}$.

When executing generated programs, we incorporate environment state feedback in response to asserts. VH provides observations in the form of state graph with object properties and relations. To check assertions in this environment, we extract information about the relevant object from the state graph and prompt the LLM to return whether the assertion holds or not given the state graph and assertion as a text prompt (Fig. 2 *Prompt for State Feedback*).

### B. Real-Robot Experiments

We use a Franka-Emika Panda robot with a parallel-jaw gripper. We assume access to a pick-and-place policy. The policy takes as input two pointclouds of a target object and a target container, and performs a pick-and-place operation to place the object on or inside the container. We use the system of [37] to implement the policy, and use MPPI for motion generation, SceneCollisionNet [37] to avoid collisions, and generate grasp poses with Contact-GraspNet [38].

We specify a single import statement for the action grab_and_putin(obj1, obj2) for PROGPROMPT. We use ViLD [39], an open-vocabulary object detection model, to identify and segment objects in the scene and construct the available object list for the prompt. Unlike in the virtual environment, where object list was a global variable in common for all tasks, here the object list is a local variable for each plan function, which allows greater flexibility to adapt to new objects. The LLM outputs a plan containing function calls of form grab_and_putin(obj1, obj2). Here, objects obj1 and obj2 are text strings that we map to pointclouds using ViLD segmentation masks and the depth image. Due to real world uncertainty, we do not implement assert-based closed loop options on the tabletop plans.

### C. Evaluation Metrics

We use three metrics to evaluate system performance: success rate (*SR*), goal conditions recall (*GCR*), and executability (*Exec*). The task-relevant goal-conditions are the set of goal-conditions that changed between the initial and final state in the demonstration. *SR* is the fraction of executions that achieved all task-relevant goal-conditions. *Exec* is the fraction of actions in the plan that are executable in the environment, even if they are not relevant for the task. *GCR* is measured using the set difference between ground truth final state conditions $\mathbf{g}$ and the final state achieved $\mathbf{g}'$ with the generated plan, divided by the number of task-specific goal-conditions; *SR*= 1 only if *GCR*= 1.

## V. RESULTS

PROGPROMPT successfully prompts LLM-based task planners to both virtual and physical agent tasks.

### A. Virtual Experiment Results

Tab. I summarizes the performance of our task plan generation and execution system in the *seen* environment of VirtualHome. We utilize a GPT3 as a language model backbone to receive PROGPROMPT prompts and generate plans. Each result is averaged over 5 runs in a single VH

environment across 10 tasks. The variability in performance across runs arises from sampling LLM output. We include 3 Pythonic task plan examples per prompt after evaluating performance on VH for between 1 prompt and 7 prompts and finding that 2 or more prompts result in roughly equal performance for GPT3. The plan examples are fixed to be: "*put the wine glass in the kitchen cabinet*", "*throw away the lime*", and "*wash mug*".

We can draw several conclusions from Tab. I. First, PROGPROMPT (rows 3-6) outperforms prior work [2] (row 8) by a substantial margin on all metrics using the same large language model backbone. Second, we observe that the CODEX [28] and DAVINCI models [27]—themselves GPT3 variants—show mixed success at the task. In particular, DAVINCI does not match base GPT3 performance (row 2 versus row 3), possibly because its prompt length constraints limit it to 2 task examples versus the 3 available to other rows. Additionally, CODEX *exceeds* GPT3 performance on every metric (row 1 versus row 3), likely because CODEX is explicitly trained on programming language data. However, CODEX has limited access in terms of number of queries per minute, so we continue to use GPT3 as our main LLM backbone in the following ablation experiments. Our recommendation to the community is to utilize a program-like prompt for LLM-based task planning and execution, for which base GPT3 works well, and we note that an LLM fine-tuned further on programming language data, such as CODEX, can do even better.

We explore several ablations of PROGPROMPT. First, we find that FEEDBACK mechanisms in the example programs, namely the `assert`ions and recovery actions, improve performance (rows 3 versus 4 and 5 versus 6) across metrics, the sole exception being that *Exec* improves a bit without FEEDBACK when there are no COMMENTS in the prompt example code. Second, we observe that removing COMMENTS from the prompt code substantially reduces performance on all metrics (rows 3 versus 5 and 4 versus 6), highlighting the usefulness of the natural language guidance within the programming language structure.

We also evaluate LANGPROMPT, an alternative to PROGPROMPT that builds prompts from natural language text description of objects available and example task plans (row 7). LANGPROMPT is similar to the prompts built by [2]. The outputs of LANGPROMPT are generated action sequences, rather than our proposed, program-like structures. Thus, we finetune GPT2 to learn a policy $P(\mathbf{a}_t|\mathbf{s}_t, \text{GPT3 step}, \mathbf{a}_{1:t-1})$ to map those generated sequences to executable actions in the simulation environment. We use the 35 tasks in the training set, and annotate the text steps and the corresponding action sequence to get 400 data points for training and validation of this policy. We find that while this method achieves reasonable partial success through *GCR*, it does not match [2] for program executability *Exec* and does not generate any fully successful task executions.

**Task-by-Task Performance** PROGPROMPT performance for each task in the test set is shown in Table II. We observe that tasks that are similar to prompt examples, such as *throw*

TABLE II: PROGPROMPT performance on the VH test-time tasks and their ground truth actions sequence lengths |A|.

| Task Desc | \|A\| | SR | Exec | GCR |
|---|---|---|---|---|
| *watch tv* | 3 | 0.20±0.40 | 0.42±0.13 | 0.63±0.28 |
| *turn off light* | 3 | 0.40±0.49 | 1.00±0.00 | 0.65±0.30 |
| *brush teeth* | 8 | 0.80±0.40 | 0.74±0.09 | 0.87±0.26 |
| *throw away apple* | 8 | 1.00±0.00 | 1.00±0.00 | 1.00±0.00 |
| *make toast* | 8 | 0.00±0.00 | 1.00±0.00 | 0.54±0.33 |
| *eat chips on the sofa* | 5 | 0.00±0.00 | 0.40±0.00 | 0.53±0.09 |
| *put salmon in the fridge* | 8 | 1.00±0.00 | 1.00±0.00 | 1.00±0.00 |
| *wash the plate* | 18 | 0.00±0.00 | 0.97±0.04 | 0.48±0.11 |
| *bring coffeepot and cupcake to the coffee table* | 8 | 0.00±0.00 | 1.00±0.00 | 0.52±0.14 |
| *microwave salmon* | 11 | 0.00±0.00 | 0.76±0.13 | 0.24±0.09 |
| Avg: $0 \le |A| \le 5$ | | 0.20±0.40 | 0.61±0.29 | 0.60±0.25 |
| Avg: $6 \le |A| \le 10$ | | 0.60±0.50 | 0.95±0.11 | 0.79±0.29 |
| Avg: $11 \le |A| \le 18$ | | 0.00±0.00 | 0.87±0.14 | 0.36±0.16 |

*away apple* versus *wash the plate* have higher *GCR* since the ground truth prompt examples hint about good stopping points. Even with high *Exec*, some task *GCR* are low, because some tasks have multiple appropriate goal states, but we only evaluate against a single "true" goal. For example, after microwaving and plating salmon, the agent may put the salmon on a table or a countertop.

TABLE III: PROGPROMPT results on Virtual Home in additional scenes. We evaluate on 10 tasks each in two additional VH scenes beyond scene ENV-0 where other reported results take place.

| VH Scene | SR | Exec | GCR |
|---|---|---|---|
| ENV-0 | 0.34±0.08 | 0.84±0.01 | 0.65±0.05 |
| ENV-1 | 0.56±0.08 | 0.85±0.02 | 0.81±0.07 |
| ENV-2 | 0.56±0.05 | 0.85±0.03 | 0.72±0.09 |
| Average | 0.48±0.13 | 0.85±0.02 | 0.73±0.10 |

**Other Environments** We evaluate PROGPROMPT in two additional VH environments (Tab. III). For each, we append a new object list representing the new environment after the example tasks in the prompt, followed by the task to be completed in the new scene. The action primitives and other PROGPROMPT settings remain unchanged. We evaluate on 10 tasks with 5 runs each. For new tasks like *wash the cutlery in dishwasher*, PROGPROMPT is able to infer that *cutlery* refers to *spoons* and *forks* in the new scenes, despite that *cutlery* always refers to *knives* in example prompts.

### B. Qualitative Analysis and Limitations

We manually inspect generated programs and their execution traces from PROGPROMPT and characterize common failure modes. Many failures stem from the decision to make PROGPROMPT *agnostic* to the deployed environment and its peculiarities, which may be resolved through explicitly communicating, for example, object affordances of the target environment as part of the PROGPROMPT prompt.

- *Environment artifacts*: the VH agent cannot find or interact with objects nearby when `sitting`, and some common sense actions for objects, such as `opening` a *tvstand*'s cabinets, are not available in VH.

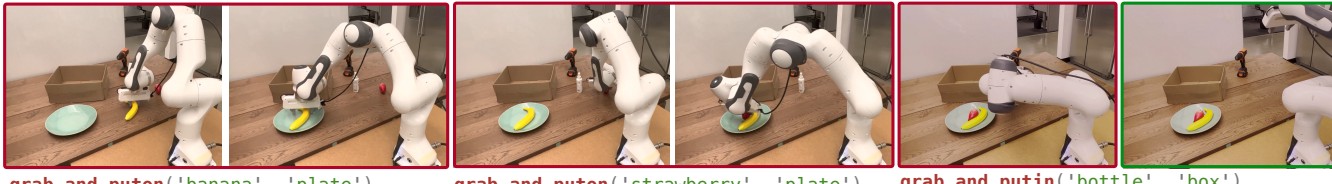

**Fig. 4:** Robot plan execution rollout example on the sorting task showing relevant objects *banana*, *strawberry*, *bottle*, *plate* and *box*, and a distractor object *drill*. The LLM recognizes that *banana* and *strawberry* are *fruits*, and generates plan steps to place them on the *plate*, while placing the *bottle* in the *box*. The LLM ignores the distractor object *drill*. See Figure 1 for the prompt structure used.

- *Environment complexities*: when an object is not accessible, the generated assertions might not be enough. For example, if the agent finds an object in a *cabinet*, it may not plan to `open` the *cabinet* to `grab` the object.
- *Action success feedback* is not provided to the agent, which may lead to failure of the subsequent actions. Assertion recovery modules in the plan can help, but aren't generated to cover all possibilities.
- *Incomplete generation:* Some plans are cut short by LLM API caps. One possibility is to query the LLM again with the prompt and partially generated plan.

In addition to these failure modes, our strict final state checking means if the agent completes the task *and some*, we may infer failure, because the environment goal state will not match our precomputed ground truth final goal state. For example, after making *coffee*, the agent may take the *coffeepot* to another *table*. Similarly, some task descriptions are ambiguous and have multiple plausible correct programs. For example, "*make dinner*" can have multiple possible solutions. PROGPROMPT generates plans that cooks *salmon* using the *fryingpan* and *stove*, and sometimes the agent adds *bellpepper* or *lime*, or sometimes with a side of *fruit*, or served in a *plate* with *cutlery*. When run in a different VH environment, the agent cooks *chicken* instead. PROGPROMPT is able to generate plans for such complex tasks as well while using the objects available in the scene and not explicitly mentioned in the task. However, automated evaluation of such tasks requires enumerating all valid and invalid possibilities or introducing human verification.

### C. *Physical Robot Results*

The physical robot results are shown in Tab. IV. We evaluate on 4 tasks of increasing difficulty listed in Tab. IV. For each task we perform two experiments: one in a scene that only contains the necessary objects, and with one to three distractor objects added.

All results shown use PROGPROMPT with comments, but not feedback. Our physical robot setup did not allow reliably tracking system state and checking `assertions`, and is prone to random failures due to things like grasps slipping. The real world introduces randomness that complicates a quantitative comparison between systems. Therefore, we intend the physical results to serve as a qualitative demonstration of the ease with which our prompting approach allows constraining and grounding LLM-generated plans to a physical robot system. We report an additional metric *Plan SR*, which refers to whether the plan would have *likely succeeded*, provided successful pick-and-place execution without gripper failures.

Across tasks, with and without distractor objects, the system almost always succeeds, failing only on the *sort* task. The run without distractors failed due to a random gripper failure. The run with 2 distractors failed because the model mistakenly considered a *soup can* to be a *bottle*. The executability for the generated plans was always *Exec*=1.

**TABLE IV:** Results on the physical robot by task type.

| Task Description | Distractors | SR | Plan SR | GCR |
|---|---|---|---|---|
| *put the banana in the bowl* | 0 | 1 | 1 | 1/1 |
| | 4 | 1 | 1 | 1/1 |
| *put the pear on the plate* | 0 | 1 | 1 | 1/1 |
| | 4 | 1 | 1 | 1/1 |
| *put the banana on the plate* | 0 | 1 | 1 | 2/2 |
| *and the pear in the bowl* | 3 | 1 | 1 | 2/2 |
| *sort the fruits on the plate* | 0 | 0 | 1 | 2/3 |
| *and the bottles in the box* | 1 | 1 | 1 | 3/3 |
| | 2 | 0 | 0 | 2/3 |

### VI. CONCLUSIONS AND FUTURE WORK

We present an LLM prompting scheme for robot task planning that brings together the two strengths of LLMs: commonsense reasoning and code understanding. We construct prompts that include situated understanding of the world and robot capabilities, enabling LLMs to directly generate executable plans as programs. Our experiments show that PROGPROMPT programming language features improve task performance across a range of metrics. Our method is intuitive and flexible, and generalizes widely to new scenes, agents and tasks, including a real-robot deployment.

As a community, we are only scratching the surface of task planning as robot plan generation and completion. We hope to study broader use of programming language features, including real-valued numbers to represent measurements, nested dictionaries to represent scene graphs, and more complex control flow. Several works from the NLP community show that LLMs can do arithmetic and understand numbers, yet their capabilities for complex robot behavior generation are still relatively under-explored.

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
