# OpenReview forum: "ProgPrompt: Generating Situated Robot Task Plans using Large Language Models"
_robot-learning.org/CoRL/2022/Workshop/LangRob — LangRob 2022 Poster_

### Official Review · Reviewer_KMfD · 2022-11-10
**Novel marriage of LLMs and task planning**

**Rating:** 7
**Confidence:** 3

**Review:**

The authors present ProgPrompt, a novel method for utilizing a LLM for planning by supplying program-like prompts and executing output programs as robot plans. What separates this research most notably from other works that use LLMs for robot task planning is that ProgPrompt utilizes programming language structures in prompts to condition the LLMs to generate pythonic programs that have grounded interaction with the environment via API calls to actions and assertions. The authors do a good job of explaining their approach and experimental methodology, which they perform on large dataset (70 hand-crafted household tasks.) The authors also introduce a number of new metrics for measuring the performance of their approach, as well as perform a qualitative analysis of the programs (robot plans) generated by the LLMs.

Overall the paper is easy to follow, though the diagrams are slightly difficult to interpret (too many annotations in Fig 2, and it’s unclear what the meaning of the arrows are). The work provides a new way to attach LLMs to robots, and is therefore original and significant.

Pros
——
- High quality of writing and explanation
- Large dataset for evaluation
- Ablation study
- Bridges NLP with robotics in a novel way

Cons
——
- New evaluation metrics make it difficult to compare this work to others
- Performance was compared only to a single baseline (Huang et al.)
- The ProgPrompt pythonic language is quite primitive and may not be able to express more complex tasks

---

### Official Review · Reviewer_JJfW · 2022-11-11
**Extensive experiments, well written study on LLMs for robotic task planning**

**Rating:** 7
**Confidence:** 4

**Review:**

**Summary**:

This work introduces ProgPrompt, a programmatic prompt format that provides a structured template for LLMs to perform semantic task planning. The main contribution is to express the prompt template in a very structured way, explicitly providing available objects and action affordances. They showcase plan proposals in the simulated Virtual Home environment as well as real world robot experiments. Finally, they study various design decisions of the prompt structure, such as feedback and recovery actions.


**Strengths**:
The paper is written well with extensive empirical results in both simulation and the real world

**Weaknesses**:
Figure 2 is fairly complicated and could improved for clarity
While it is concurrent work, it would be useful to compare with the recent https://arxiv.org/abs/2209.07753, which also employs a structured programmatic approach to prompting for robotic planning and control.

---

### Decision · Program_Chairs · 2022-11-15

Accept (Poster)